# Clustering Gene Expressions Using the Table Invitation Prior

**DOI:** 10.3390/genes13112036

**Published:** 2022-11-04

**Authors:** Charles W. Harrison, Qing He, Hsin-Hsiung Huang

**Affiliations:** Department of Statistics and Data Science, University of Central Florida, 4000 Central Florida Blvd, Orlando, FL 32816, USA

**Keywords:** Bayesian clustering, distance-dependent clustering, genome-wide association studies, gene expression

## Abstract

A prior for Bayesian nonparametric clustering called the Table Invitation Prior (TIP) is used to cluster gene expression data. TIP uses information concerning the pairwise distances between subjects (e.g., gene expression samples) and automatically estimates the number of clusters. TIP’s hyperparameters are estimated using a univariate multiple change point detection algorithm with respect to the subject distances, and thus TIP does not require an analyst’s intervention for estimating hyperparameters. A Gibbs sampling algorithm is provided, and TIP is used in conjunction with a Normal-Inverse-Wishart likelihood to cluster 801 gene expression samples, each of which belongs to one of five different types of cancer.

## 1. Introduction

The goal of clustering is to provide informative groupings (i.e., clusters) of similar objects. The objects are referred to in this paper as “subjects”, and an example of an individual subject is a single gene expression from one person. The term “subjects” will be used throughout this paper in order to avoid confusion associated with the term “samples” in a statistical context. Note that, in practice, an individual subject may correspond to an individual vector, matrix, or higher-order tensor. In this work, vectors are considered for the sake of simplicity. In contrast, a “subject index” refers to an identifier for a subject. For example, in vector-variate data, a subject index i∈{1,2,…,n} refers to the *i*th row in the provided dataset and *n* is the total number of subjects. The notation $x_{i}$ is used to refer to the subject (object) itself (i.e., a vector, matrix, tensor, etc.). The goal of Bayesian clustering is to produce a set of cluster assignments for each subject while also calculating the probability that two subjects are clustered together given the observed data and prior assumptions. Mathematically, this is represented as
(1)P(c∣X)∝P(X∣c)P(c)
where X refers to the data, c is a vector of *n* cluster assignments (e.g., the cluster assignments for each of the *n* gene expression samples), and P(c∣X) represents the posterior probability of a cluster configuration, P(X∣c) is the likelihood, and P(c) is the cluster prior which is the focus of this paper.

A well-known challenge in clustering is to estimate the unknown number of clusters $K^{*}$ [1,2]. Some clustering methods, such as MCLUST, involve an analyst fitting several models with varying degrees of complexity and then choosing the desired model based on a chosen clustering metric [3,4]. A distinct, though similar approach, uses the gap statistic [2] in conjunction with another clustering algorithm (e.g., hierarchical clustering) to estimate the number of clusters.

Bayesian nonparametric models refer to a flexible class of prior distributions that may be used in a variety of settings including clustering. In the context of clustering, the use of such priors means that an analyst does not need to specify an estimate for the number of clusters since the number of clusters is modeled as a random variable. A variety of methods have been proposed to accomplish this task, and the relevant methods are reviewed in the sections that follow. Two methods include the Ewens-Pitman Attraction (EPA) prior [5] and a well-known special case of EPA called the Chinese Restaurant Process (CRP) [6,7].

The CRP, a variant of the Dirichlet process, is a well-known prior used in Bayesian clustering. One drawback of CRP is that it does not utilize information pertaining to the similarity between subjects (e.g., gene expression samples). A natural extension of the CRP is one that includes the aforementioned similarity information, and one such extension is the EPA prior. Although EPA utilizes similarity information, the primary drawback of EPA is that it relies on the choice of a hyperparameter (this is α in Section 1.2 below). Consequently, an analyst using EPA must either choose a fixed value for the EPA hyperparameter α or rely on an approximate posterior distribution for α [5,8]. In the context of Bayesian clustering, a number of samples are taken from a posterior distribution which can be time consuming, so manually tuning a hyperparameter is not desirable.

The focus of this paper is the Table Invitation Prior (TIP) which is an attempt to maintain the advantage of Bayesian clustering (i.e., the analyst does not have to specify the number of clusters) while removing the need for an analyst to carefully tune a hyperparameter. Although the approximate posterior distribution for α used in EPA removes the need of the analyst to tune the hyperparameter, empirical results show that TIP gives superior results and is less susceptible to splitting clusters as compared to EPA. Bayesian clustering methods often rely on the use of similarity functions to capture the relationships between subjects (e.g., gene expression samples), and thus a brief review of pairwise similarity functions is provided.

### 1.1. Pairwise Similarity Functions

Some Bayesian clustering priors use the similarity between subjects in order to obtain clusters that contain subjects that are similar to each other [5,9]. Let the similarity between two subjects with indices *i* and *j* be given by λ(i,j) for i=1,2,…,n and j=1,2,…,n where *n* is the number of subjects. The similarity function λ may take a variety of forms, and in this paper the similarity function used is the exponential decay function [5,9]:(2)$$\lambda \left(i,j\right)=\exp\left(-\tau d_{ij}\right)$$
where τ>0 is a hyperparameter and $d_{ij}$ is the distance between the *i*th and *j*th subjects. Following the approach taken in [10], the hyperparameter τ is set to the following:(3)$$\hat{\tau }=\frac{1}{\tilde{d}}$$
where $\tilde{d}$ is the median of the pairwise distances of the strictly upper triangular portion of the distance matrix:(4)$$\tilde{d}=\mathrm{median}\left\{d_{ij}:i>j,i,j\in \left\{1,2,\ldots ,n\right\}\right\}.$$

The choice of the median is heuristic, but there is a justification. Equation (Equation 3) implies that
$$\lim_{d_{ij}\overset{}{\to }\infty }\exp\left(\frac{-d_{ij}}{\tilde{d}}\right)=0,$$
$$\lim_{d_{ij}\overset{}{\to }0}\exp\left(\frac{-d_{ij}}{\tilde{d}}\right)=1,$$
and
$$\lim_{d_{ij}\overset{}{\to }\tilde{d}}\exp\left(\frac{-d_{ij}}{\tilde{d}}\right)=\exp\left(-1\right).$$
Consequently, similarity values corresponding to subject pairs whose distances are significantly larger than the overall median distance go to zero whereas subject pairs that are very close to each other have a similarity value that is closer to 1. Subject pairs whose distance from each other is close to the overall median distance have a similarity value that is between 0 and 1.

### 1.2. Ewens-Pitman Attraction Prior

The EPA distribution uses the pairwise similarity between subjects and a sequential sampling scheme to induce a partition of *n* subjects [5,11]. Let $\sigma =\{\sigma_{1},\sigma_{2},\ldots ,\sigma_{n}\}$ be a random permutation of the subject indices {1,2,…,n}. Then the conditional probability of a subject with index *i* joining cluster *k* is given by the following:(5)$$P\left(c_{\sigma_{i}}=k|\alpha ,\delta ,\lambda ,c\left(\sigma_{1},\ldots ,\sigma_{i-1}\right)\right)=\left\{\begin{array}{cc} \frac{i-1-\delta q_{i-1}}{\alpha +i-1}\frac{\sum_{\sigma_{s}\in S}\lambda \left(\sigma_{i},\sigma_{s}\right)}{\sum_{s=1}^{i-1}\lambda \left(\sigma_{i},\sigma_{s}\right)}\; & \mathrm{if}\;S\in c(\sigma_{1},\ldots ,\sigma_{i-1})\\ \frac{\alpha +\delta q_{i-1}}{\alpha +i-1} & \mathrm{if}\;S\;\mathrm{is}\;a\;\mathrm{new}\;\mathrm{cluster} \end{array}\right.$$
where α>0 is a hyperparameter that controls the extent to which a new cluster is created, $q_{i-1}$ is the number of clusters that are assigned among the first i−1 subjects, δ∈[0,1) is a “discount” hyperparameter, λ is a similarity function, and $c(\sigma_{1},\sigma_{2},\ldots ,\sigma_{i-1})$ are the part assignments for the first i−1 permuted subjects $\sigma_{1},\ldots ,\sigma_{i-1}$. The discount parameter is specific to EPA and its purpose is to incorporate information about the number of clusters in a previous iteration when computing the probability of a new cluster in the current iteration.

The value for α may be treated as a constant or it can be sampled from a distribution as described in West [8]. Specifically, West’s approximate posterior distribution for α, given the number of clusters $n_{k}$, is:(6)$$\alpha |n_{k}∼\Gamma \left(a+n_{k}-1,b+\gamma +log\left(n\right)\right)$$
where Γ denotes the gamma distribution, γ is Euler’s constant, and the prior parameters are *a* and *b*. In this work, a=b=1 so that the prior for α has exponential distribution with a scale parameter of 1.

#### Chinese Restaurant Process

The CRP is a special case of EPA that occurs when the discount parameter δ=0 and λ(i,j) is constant for all subject indices *i* and *j* [5,6,7]. The conditional probability of a subject $x_{i}$ joining cluster *k* is given by the following:(7)$$P\left(c_{\sigma_{i}}=k|\alpha ,c\left(\sigma_{1},\sigma_{2},\ldots ,\sigma_{i-1}\right)\right)=\left\{\begin{array}{cc} \frac{|S|}{\alpha +i-1} & \mathrm{if}\;S\in c(\sigma_{1},\sigma_{2},\ldots ,\sigma_{i-1})\\ \frac{\alpha }{\alpha +i-1} & \mathrm{if}\;S\;\mathrm{is}\;a\;\mathrm{new}\;\mathrm{subset} \end{array}\right.$$

The CRP is obtained by taking the product of (Equation 7) over all possible partitions.

## 2. Table Invitation Prior

In this section, the Table Invitation Prior (TIP) is presented in the context of a Gibbs sampler in iteration t=1,2,…,T. An analogy is now provided to illustrate the prior’s mechanics. Suppose that *n* subjects $x_{1},x_{2},\ldots ,x_{n}$ (i.e., vectors, matrices, tensors, etc.) are sitting in a restaurant with $k=1,2,\ldots ,K^{\left(t\right)}$ tables (clusters). A randomly selected subject with index *r* is chosen and then the ${\hat{n}}_{\tau_{r}}$ subjects that are most similar to the subject with index *r* are “invited” to a new table (cluster) $K^{\left(t\right)}+1$ (in this paper, all the ${\hat{n}}_{\tau_{r}}$ subjects accept the invitation with probability 1). The posterior probability of every subject belonging to a table (cluster) is computed for tables (clusters) $1,2,\ldots ,K^{\left(t\right)},K^{\left(t\right)}+1$ and, using the probabilities, the subjects are randomly assigned to a table (i.e., sample the posterior cluster assignment for every subject). The variable *t* is incremented by 1 and the this process continues; here *t* is the number of times the above process has occurred so far.

A more formal description of the Table Invitation Prior now follows. For the iteration *t* in a Gibbs sampler, let the random variable *r* be a randomly selected subject index (e.g., a randomly selected index corresponding to an individual gene expression) from a discrete uniform distribution
(8)r∼U{0,n}
so that r∈{1,2,…,n}. Suppose a random subject $x_{r}$ is selected (i.e., $x_{r}$ can be a vector, matrix, higher-order tensor, etc.). The set of similarity values between subject $x_{r}$, itself, and the other n−1 subjects is
(9)$$\Lambda_{r}=\left\{\lambda \left(r,i\right):i\in \left\{1,2,\ldots ,n\right\}\right\}$$
where λ(r,i) is the similarity between the *r*th subject and the *i*th subject. Let the *j*th ordered similarity value in the set $\Lambda_{r}$ be $\Lambda_{r^{\left(j\right)}}$ for j=1,2,…,n and let
(10)$$\Lambda_{r^{\left(n\right)}}=\lambda \left(r,r\right)>\Lambda_{r^{\left(n-1\right)}}>\Lambda_{r^{\left(n-2\right)}}\ldots >\Lambda_{r^{\left(1\right)}}.$$
The set of indices of the $n_{\tau_{r}}$ subjects that are most similar to subject $x_{r}$ is given by
(11)$$S_{r}=\left\{r=r^{\left(n\right)},r^{\left(n-1\right)},r^{\left(n-2\right)},\ldots ,r^{\left(n-n_{\tau_{r}}+1\right)}\right\}$$
where $n_{\tau_{r}}\in \left\{1,2,\ldots ,n-1\right\}$ is a hyperparameter. The estimation of hyperparameter $n_{\tau_{r}}$ proceeds in the following manner. First, recall that *r* is a randomly selected subject index so that r∈{1,2,…,n}. The pairwise distances with respect to subject *r* are extracted and the distance from subject *r* to itself is removed:$$d_{r}=\left\{d_{r,j}:j\in \left\{1,2,\ldots ,r-1,r+1.\ldots ,n\right\}\right\},$$
The distances are then sorted in ascending order:(12)$$d_{r}^{*}=\left\{d_{r,j^{*}}:d_{r,j^{*}}<d_{r,j^{*}+1}\;,j^{*}\in \left\{1,2,\ldots ,r-1,r+1,\ldots ,n\right\}\right\}.$$

Next, a univariate multiple change point detection algorithm is applied to the sorted distances. The change point detection algorithm used in this paper is binary segmentation from the changepoint library in R [12]. The binary segmentation function in R takes a hyperparameter, denoted by *Q*, that is the maximum number of changepoints (Equation 13). In this paper, the value is set to $\left\lfloor \frac{n}{2}+1\right\rfloor$ since the changepoint library will throw an error if $Q>\frac{n}{2}+1$; also this allows the change point method to have the maximum amount of flexibility in detecting changes in the subject distances. Let the set of change points be given by
(13)$${\hat{\tau }}_{r}=\left\{{\hat{\tau }}_{r,1},{\hat{\tau }}_{r,2},\ldots \right\},$$
then the number of subjects that are similar to the subject with index *r* is taken to follow a Poisson distribution:$${\hat{n}}_{\tau_{r}}∼Poi\left({\hat{\tau }}_{r,1}\right).$$
Consequently, the set ${\hat{S}}_{r}$ is given by:(14)$${\hat{S}}_{r}=\left\{r=r^{\left(n\right)},r^{\left(n-1\right)},r^{\left(n-2\right)},\ldots ,r^{\left(n-{\hat{n}}_{\tau_{r}}+1\right)}\right\}.$$
Let the vector containing the cluster assignments be denoted by $c_{t}$ so that the *i*th element contains the cluster assignment for the *i*th subject and
$$c_{t,i}\in \left\{1,2,\ldots ,K^{\left(t\right)}\right\}\;\mathrm{for}\;i=1,2\ldots ,n$$
where $K^{\left(t\right)}$ is the total number of clusters after posterior sampling in iteration *t*. The Table Invitation Prior is based on selecting a random subject index *r* and forming a new cluster with ${\hat{n}}_{\tau_{r}}$ subjects that are most similar to subject $x_{r}$. That is, the new cluster is formed using the subjects whose indices are in the set ${\hat{S}}_{r}$. Consider a modified cluster vector ${\tilde{c}}_{t}$
(15)$${\tilde{c}}_{t,i}=\left\{\begin{array}{cc} c_{t,i} & i\notin {\hat{S}}_{r}\\ K^{\left(t\right)}+1 & i\in {\hat{S}}_{r}, \end{array}\right.$$
then the Table Invitation Prior (TIP) is given by
(16)$$P\left(c_{t+1,i}=k|X,p,\lambda \right)\propto \sum_{{\tilde{c}}_{t,j}=k}\lambda \left(i,j\right)\;\mathrm{for}\;k\in \left\{1,2,\ldots ,K^{\left(t\right)},K^{\left(t\right)}+1\right\}.$$

## 3. TIP Gibbs Sampler

An implementation of a Gibbs sampler with a Table Invitation Prior for clustering corresponds to the following steps, and it is summarized in Algorithm 1. Initially, all subjects are sitting at $K^{\left(0\right)}$ tables (clusters). For Gibbs sampling iteration t=1, a random subject with index *r* is chosen and the ${\hat{n}}_{\tau_{r}}$ most similar subjects with indices $r_{\left(n-1\right)},r_{\left(n-2\right)},\ldots ,r_{\left(n-{\hat{n}}_{\tau_{r}}+1\right)}$ are assigned to table $K^{\left(0\right)}+1$ with subject $x_{r}$. The conditional probabilities for all subjects $x_{1},x_{2},\ldots ,x_{n}$ (i.e. vectors, matrices, tensors, etc.) are computed using Equation (Equation 16) for all clusters $k=1,2,\ldots ,K^{\left(0\right)}+1$; if desired, a likelihood value may be computed for each table (cluster). Next, the posterior probability is computed and the subject’s posterior cluster assignment is sampled (i.e., sampled from the set $\left\{1,2,\ldots ,K^{\left(0\right)},K^{\left(0\right)}+1\right\}$). This gives a partition with $K^{\left(1\right)}$ clusters (tables). In the second Gibbs sampling iteration t=2, a random subject with index r∼U{0,n} is chosen and the ${\hat{n}}_{\tau_{r}}$ most similar subjects with indices $r_{\left(n-1\right)},r_{\left(n-2\right)},\ldots ,r_{\left(n-{\hat{n}}_{\tau_{r}}+1\right)}$ are assigned to table $K^{\left(1\right)}+1$ with subject $x_{r}$. The conditional probabilities for all subjects $x_{1},x_{2},\ldots ,x_{n}$ are computed using Equation (Equation 16) for all clusters $k=1,2,\ldots ,K^{\left(1\right)}+1$; again, a likelihood value may be computed for each table (cluster), and each subject’s posterior cluster assignment is sampled (i.e., sampled from the set $\left\{1,2,\ldots ,K^{\left(1\right)}+1\right\}$). This process continues for t∈{3,…,T}.
**Algorithm 1:** Table Invitation Prior Clustering
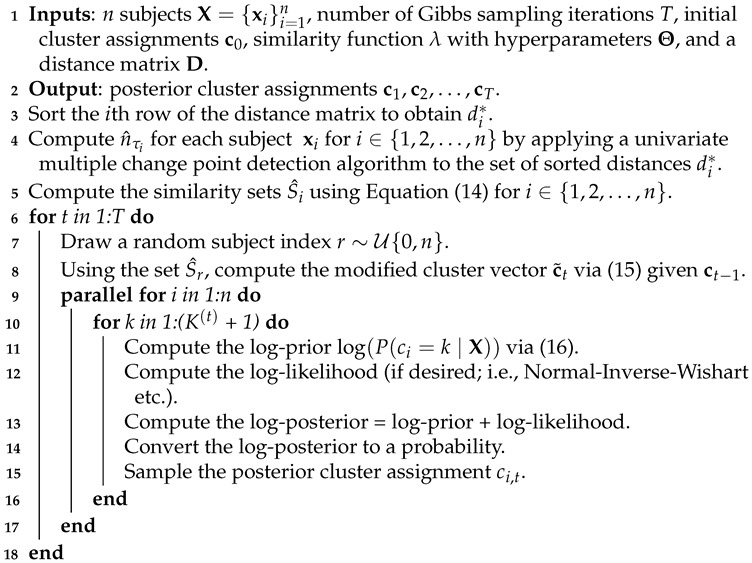


### 3.1. Posterior Cluster Assignments

The TIP Gibbs sampler produces a set of posterior cluster vectors $c_{1},c_{2},\ldots ,c_{T}$. However, it is necessary to produce a single clustering from this set of posterior cluster assignments, and this section describes the methodology used in this paper to accomplish this task (other methods may be used for this task). Each posterior cluster vector is transformed into an n×n posterior proximity matrix $B^{\left(t\right)}$:(17)$$B_{i,j}^{\left(t\right)}=\left\{\begin{array}{cc} 0 & c_{t,i}\neq c_{t,i}\\ 1 & c_{t,i}=c_{t,j} \end{array}\right.$$
where $c_{t,i}$ and $c_{t,j}$ are the posterior cluster assignments for subject $x_{i}$ and $x_{j}$ after Gibbs sampling iteration t∈{1,2,…,T}. The posterior similarity matrix is given by
(18)$$\bar{B}=\frac{1}{T}\sum_{t=1}^{T}B^{\left(t\right)}.$$

A vector of posterior cluster assignments is computed using the Posterior Expected Adjusted Rand (PEAR) index, and technical details as well as an application of PEAR to gene expression data may be found in Fritsch and Ickstadt [13]. Let ρ denote the PEAR index function. Using the posterior similarity matrix $\bar{B}$, the cluster vector that maximizes the PEAR index is taken to be the posterior cluster assignment vector:(19)$$c^{*}=\underset{c\in \{c_{1},c_{2},\ldots ,c_{T}\}}{arg max}\rho \left(c|\bar{B}\right).$$
The computation of the PEAR index is accomplished using the mcclust package in R [13,14].

### 3.2. Likelihood Function

The Table Invitation Prior may be used for clustering vectors, matrices, and higher-order tensors, assuming that a suitable distance metric is available. One component of a Gibbs sampler utilizing TIP that may change depending on the dataset is the likelihood function. In this paper vector-variate data are considered, and the conjugate Normal-Inverse-Wishart (NIW) prior for the mean and covariance is utilized. Let $x_{i}\in R^{p}$ be a p×1 vector and represent the *i*th subject for i=1,2,…,n. Let $c_{i}$ be the cluster assignment for subject *i*. Assume that
(20)$$x_{i}|c_{i}=k∼N_{p}\left(\mu_{k},\Sigma_{k}\right),$$
(21)$$\left(\mu_{k},\Sigma_{k}\right)∼NIW\left(\mu_{0},\lambda_{0},\Psi_{0},\nu_{0}\right),$$
then the joint posterior for $\left(\mu_{k},\Sigma_{k}|x_{i}\right)$ is given by
(22)$$\left(\mu_{k},\Sigma_{k}|y_{i}\right)∼NIW\left(\mu_{1k},\lambda_{1k},\Psi_{1k},\nu_{1k}\right)$$
where NIW denotes the Normal-Inverse-Wishart distribution. The posterior arguments are given by
$$\mu_{1k}=\frac{\lambda_{0}\mu_{0}+n_{k}\bar{x}}{\lambda_{0}+n_{k}},$$
$$\lambda_{1k}=\lambda_{0}+n_{k},$$
$$\nu_{1k}=\nu_{0}+n_{k},$$
and
$$\Psi_{1k}=\Psi_{0}+\sum_{i=1}^{n_{k}}{\left(x_{i}-{\bar{x}}_{k}\right)}^{\prime }\left(x_{i}-{\bar{x}}_{k}\right)+\frac{\lambda_{0}n_{k}}{\lambda_{0}+n_{k}}\left({\bar{x}}_{k}-\mu_{0}\right){\left({\bar{x}}_{k}-\mu_{0}\right)}^{\prime }.$$

There are four hyperparameters and the following values are used:$${\hat{\mu }}_{0}=\frac{1}{n}\sum_{i=1}^{n}x_{i},$$
$${\hat{\lambda }}_{0}=1,$$
$${\hat{\nu }}_{0}=p,$$
and
$${\hat{\Psi }}_{0}={\left(\left(p-1\right)\sum_{i=1}^{n}{\left(x_{i}-\bar{x}\right)}^{\prime }\left(x_{i}-\bar{x}\right)\right)}^{-1}.$$

### 3.3. Visualizing The Posterior Similarity Matrix

The n×n symmetric matrix $\bar{B}$ can be viewed an undirected weighted graph with *n* vertices where the edge between the *i*th and *j*th subjects (vertices) represents the posterior probability that subject $x_{i}$ and subject $x_{j}$ are in the same cluster. A network plot may be used to view $\bar{B}$ directly, but the number of edges corresponding to small posterior probabilities may unnecessarily complicate the plot. Consequently, we show the plot of the graph ${\bar{B}}_{1}$ which is the result of removing the maximum number of edges in the graph $\bar{B}$ such that the number of components in the graph is 1. That is, the graph ${\bar{B}}_{1}$ has the minimum entropy among all subgraphs with one component and we call its corresponding network plot the “one-cluster plot”. The idea is to remove as many connections as possible while still maintaining one component so that the clusters’ relationships with each other are revealed. The network plots are used in Section 5 to visualize the cluster results.

## 4. Simulation Data

In this section, a clustering simulation is presented to compare TIP with various clustering algorithms including EPA, MCLUST, and linkage-based methods. For EPA, δ=0 and α follows West’s posterior given by Equation (Equation 6).

### 4.1. Simulation Description

The simulation is given by the following. A dataset X with *n* subjects $x_{1},x_{2},\ldots ,x_{n}$ is generated where $x_{n}\in R^{p}$. Each subject $x_{i}$ for i=1,2,…,n is generated according to its true cluster assignment *k* so that
$$x_{i}|k∼N_{p}\left(\mu_{k},\Sigma_{k}\right),\;\mathrm{for}\;i=1,2,\ldots ,n,$$
$$\mu_{k}∼N_{p}\left(0,10I_{p}\right),$$
and
$$\Sigma_{k}∼W^{-1}\left(I_{p},p+1\right).$$
Here $N_{p}$ denotes the *p*-variate multivariate normal distribution and $W_{p}^{-1}$ denotes the inverse Wishart distribution. The number of burn-in iterations for both TIP and EPA is set to 1000, and the number of sampling iterations is set to 1000.

### 4.2. Simulation Results: Normal-Inverse-Wishart Likelihood Function

In this section, simulation results for TIP and EPA in conjunction with a Normal-Inverse-Wishart likelihood function are presented. TIP and EPA are compared with the MCLUST algorithm, k-means clustering, and hierarchical clustering [4,15]. For k-means and hierarhcial clustering, the number of clusters is estimated using the gap statistic [2].

#### 4.2.1. $Well\;Separated\;Clusters:p=2,K^{*}=4$ and n=110

In this simulation there are $K^{*}=4$ well separated clusters. Each cluster is composed of $n_{1}=20$, $n_{2}=25$, $n_{3}=30$, and $n_{4}=35$ vectors in p=2 dimensional space. The results are shown in Figure 1. Both TIP and MCLUST cluster the datasets perfectly while EPA with West’s posterior is too aggressive and results in 12 clusters. Hierarchical clustering (complete linkage) is utilized in conjunction with the gap statistic. The optimal number of clusters given by the gap statistic is 4, and hierarchical clustering using complete linkage with exactly 4 clusters perfectly separates the data. The gap statistic is also used with k-means and the optimal number of clusters given by the gap statistic is 4, and k-means perfectly separates the data.

#### 4.2.2. Overlapped Clusters: $p=2,\;K^{*}=4$ and n=120

In this simulation there are $K^{*}=4$ clusters, but two of the clusters are overlapped. In this case, the cluster sizes are given by $n_{1}=20$, $n_{2}=25$, $n_{3}=30$, and $n_{4}=45$ vectors in p=2 dimensional space. The results are shown in Figure 2. EPA gives 15 clusters, TIP gives 11 clusters, and MCLUST gives 5 clusters. MCLUST divides the two overlapped clusters into 3 clusters whereas TIP is too aggressive and divides the overlapped clusters into 8 clusters. Similarly, EPA divides the overlapped clusters into 8 clusters, but, unlike TIP, EPA also divides Cluster 1 into 2 clusters. Hierarchical clustering (complete linkage) is used in conjunction with the gap statistic and gives 4 clusters, though the resulting cluster assignments are not necessarily accurate since part of Cluster 3 is clustered with part of Cluster 4. K-means is also applied to the dataset in conjunction with the gap statistic; the optimal number of clusters according to the gap statistic is 3 which fuses two of the true clusters (i.e., Cluster 3 and Cluster 4) together.

## 5. Application: Clustering Gene Expression Data

In this section TIP is applied to a dataset pertaining to RNA-Seq gene expression levels as measured by an Illumina HiSeq platform [16]. The data were accessed from the UCI Machine Learning Repository [17] and were collected as a part of the Cancer Genome Atlas Pan-Cancer analysis project [18]. There are n=801 gene expression samples (i.e., n=801 subjects) and p=20,531 gene expression levels. The 801 gene expression samples can be classified into one of 5 classes, and each class corresponds to a different type of cancer: BRCA, COAD, KIRC, LUAD, and PRAD.

Principal components analysis was applied to the data, and 7 principal components were used so that p=7. A plot showing the cumulative variance explained by a given number of principal components is shown in Figure 3. The reason that 7 principal components were used is that it takes a relatively large number of dimensions to explain percentages of the variance greater than 80%. The first 7 principal components explain about 80% of the variance, but it takes 22 principal components to explain 85% of the variance, 82 principal components to explain 90% of the variance, 269 principal components to explain 95% of the variance, and 603 principal components to explain 99% of the variance.

The clustering methods are applied to the principal components so that p=7 and n=801. The TIP posterior cluster assignments are shown in Table 1. There is a small overlap of classes in cluster 3 where there are 270 BRCA gene expression samples and 1 LUAD gene expression sample. Also, 30 BRCA gene expression samples form a distinct cluster (see cluster 5). The one-cluster plot is shown in Figure 4 and shows a small amount of overlap between LUAD and BRCA which is consistent with the posterior cluster assignments in Table 1.

The posterior cluster assignments for EPA are shown in Table 2. EPA is able to separate the classes quite well, but there is one cluster where there is substantial overlap between classes. Cluster 10 is comprised of samples from BRCA, COAD, KIRC, LUAD, and PRAD whereas this does not occur for TIP and MCLUST. Furthermore, Cluster 6 contains samples from both BRCA and LUAD; this is true for TIP and EPA. The one-cluster plot for EPA is shown in Figure 5, and it shows that there is overlap between LUAD and BRCA as well as BRCA and COAD.

The cluster assignments for MCLUST are shown in Table 3. MCLUST, like TIP, performs well. MCLUST produces one cluster with a minor amount of overlap: cluster 1 features 57 BRCA samples and 2 LUAD samples. Furthermore, BRCA is split between two clusters: one with 57 BRCA samples and another with 243 BRCA samples. This is similar to the TIP results.

Hierarchical clustering is applied in conjunction with the gap statistic to choose the number of clusters, and the R package cluster is used to compute the gap statistic [19]. The settings used for hierarchical clustering and k-means are the default settings in the stats library in R [20]. The results for hierarchical clustering using complete linkage are shown in Table 4. The optimal number of clusters estimated via the gap statistic is 7, but complete linkage clustering is unable to separate the classes. The results for hierarchical clustering using single linkage are shown in Table 5. The optimal number of clusters estimated by the gap statistic is 1, and thus single linkage clustering is unable to separate the classes. The results for hierarchical clustering using median linkage are shown in Table 6. The optimal number of clusters given by the gap statistic is 1, and thus median linkage clustering is unable to separate the classes. K-means clustering is also used in conjunction with the gap statistic. The optimal number of clusters according to the gap statistic is 5, but the resulting clusters, which are provided in Table 7, do not separate the data well.

## 6. Conclusions and Discussion

In this work, a Bayesian nonparametric clustering prior called the Table Invitation Prior (TIP) was introduced. TIP does not require the analyst to specify the number of clusters, and its hyperparameters are automatically estimated via univariate multiple change point detection. EPA is a prior on partitions and is used for Bayesian clustering. Unlike TIP, the probability of a new cluster in EPA depends on preset hyperparameters (i.e., δ and α>−δ), which is not data-driven, and it may lead to a bias of the number of clusters due to improper hyperparameter values. The main difference between TIP and MCLUST is that TIP is a Bayesian cluster prior which can be incorporated with various types of likelihoods and priors for the parameters in the likelihood. For example, TIP can work with a conjugate using the Normal-Inverse-Wishart prior of for unknown mean and covariance matrix. MCLUST is based on a mixture model of finite Gaussian likelihoods and uses an expectation–maximization (EM) algorithm [21] for the Gaussian mixture parameter estimation with a preset covariance structure. TIP was used in conjunction with a Normal-Inverse-Wishart conjugate prior to cluster gene expression data, and it was compared with a variety of other clustering methodologies, including another Bayesian nonparametric clustering method called EPA, MCLUST, hierarchical clustering in conjunction with the gap statistic, and k-means clustering in conjunction with the gap statistic.

## Figures and Tables

**Figure 1 genes-13-02036-f001:**
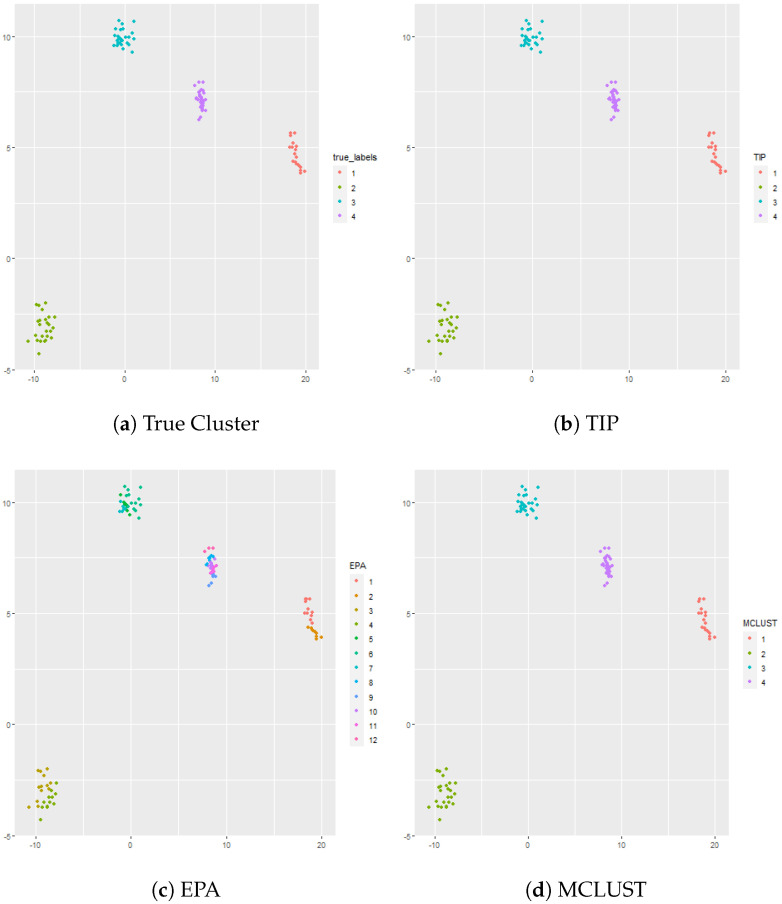
Panel (**a**) shows the true cluster assignments when $p=2,K^{*}=4$, and n=110. Panel (**b**) shows the posterior TIP cluster assignments, Panel (**c**) shows the posterior EPA assignments, Panel (**d**) shows the MCLUST assignments, Panel (**e**) shows the hierarchical clustering assignments (complete linkage) where the number of clusters is determined via the gap statistic, and Panel (**f**) shows the k-means clustering assignments where the number of clusters is determined via the gap statistic.

**Figure 2 genes-13-02036-f002:**
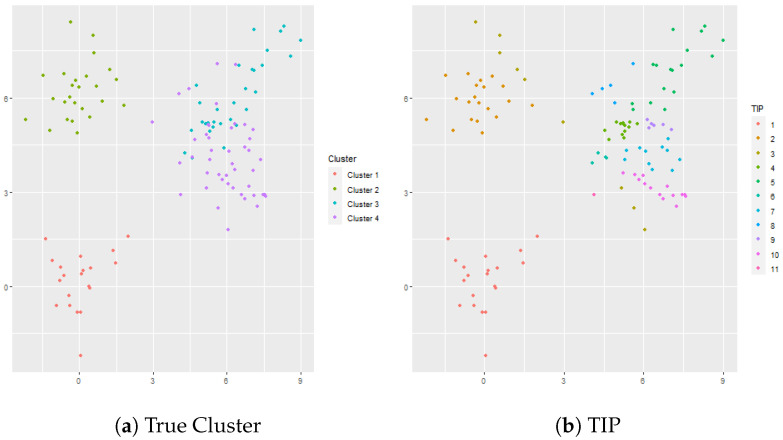
Panel (**a**) shows the true cluster assignments when $p=2,K^{*}=4$, and n=110. Panel (**b**) shows the posterior TIP cluster assignments, Panel (**c**) shows the posterior EPA assignments, Panel (**d**) shows the MCLUST assignments, Panel (**e**) shows the hierarchical clustering assignments where the number of clusters is determined via the gap statistic, and Panel (**f**) shows the k-means clustering assignments where the number of clusters is determined via the gap statistic.

**Figure 3 genes-13-02036-f003:**
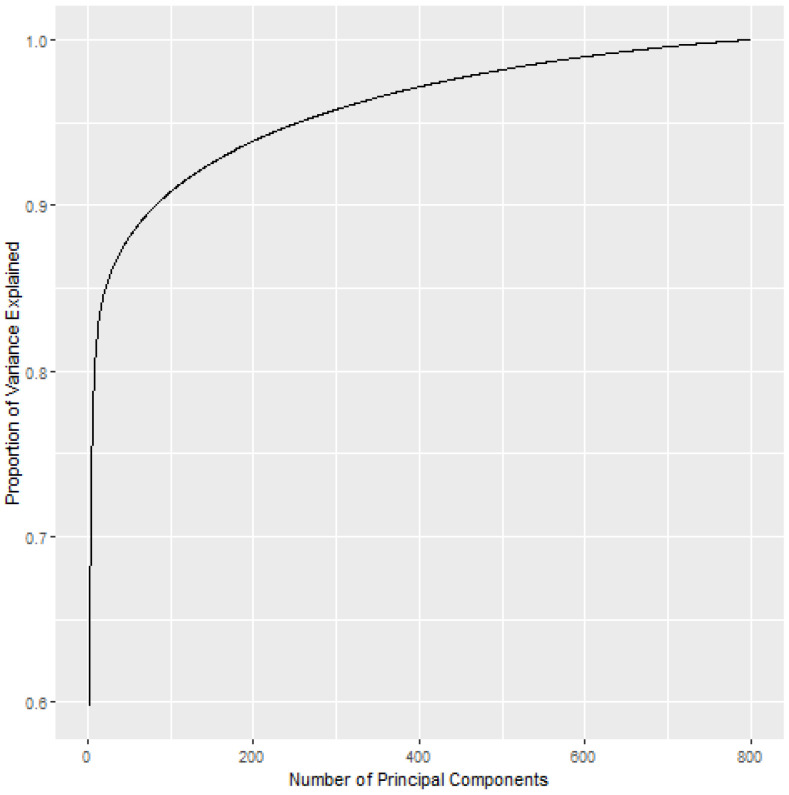
The cumulative proportion of variance explained by the principal component analysis.

**Figure 4 genes-13-02036-f004:**
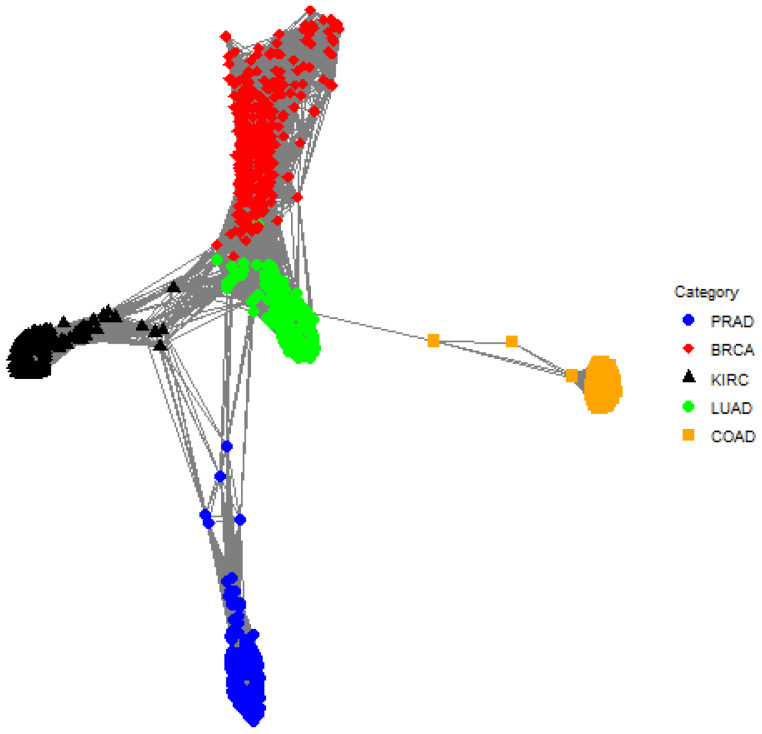
A one-cluster plot with respect to TIP where each graph vertex (i.e., a colored dot) corresponds to a subject (e.g., an individual gene expression sample) and the edge weights (i.e., the lines) correspond to the elements in the matrix ${\bar{B}}_{1}$. Specifically, the edge between subject *i* and subject *j* is the posterior probability that subject *i* and subject *j* are in the same cluster. Shorter lines correspond to larger posterior probabilities, so pairs of graph vertices that are closer to each other in the plot are more likely to be assigned to the same cluster. The plot shows a minor overlap between BRCA (red diamond) and LUAD (green circle).

**Figure 5 genes-13-02036-f005:**
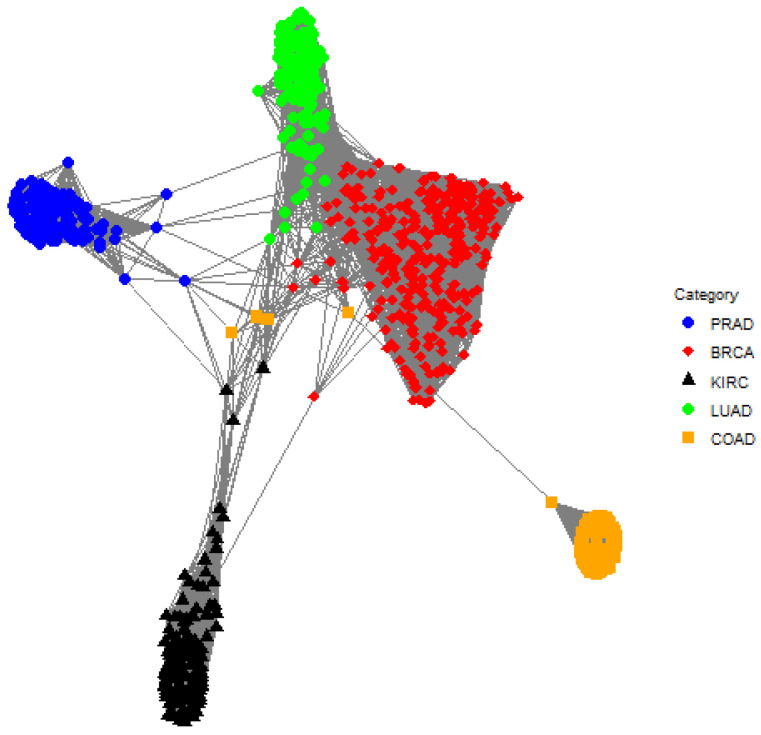
A one-cluster plot with respect to EPA where each graph vertex (i.e., a colored dot) corresponds to a subject (e.g., an individual gene expression sample) and the edge weights (i.e., the lines) correspond to the elements in the matrix ${\bar{B}}_{1}$. Specifically, the edge between subject *i* and subject *j* is the posterior probability that subject *i* and subject *j* are in the same cluster. Shorter lines correspond to larger posterior probabilities, so pairs of graph vertices that are closer to each other in the plot are more likely to be assigned to the same cluster. There is an overlap between LUAD (green circle) and BRCA (red diamond) as well as BRCA (red diamond) and COAD (orange square).

**Table 1 genes-13-02036-t001:** The distribution of the posterior TIP cluster assignments. The number in parenthesis is the number of subjects (e.g., gene expression samples) for one of the five cancer types.

Cluster ID	Distribution
1	PRAD (136)
2	LUAD (140)
3	BRCA (270) LUAD (1)
4	KIRC (146)
5	BRCA (30)
6	COAD (78)

**Table 2 genes-13-02036-t002:** The distribution of the EPA cluster assignments. The number in parenthesis is the number of subjects for one of the five cancer types.

Cluster ID	Distribution
1	PRAD (124)
2	LUAD (124)
3	PRAD (11)
4	BRCA (100)
5	KIRC (128)
6	BRCA (104) LUAD (2)
7	BRCA (32)
8	KIRC (16)
9	COAD (74)
10	BRCA (3) COAD (4) KIRC (2) LUAD (15) PRAD (1)
11	BRCA (61)

**Table 3 genes-13-02036-t003:** The distribution of the MCLUST cluster assignments. The number in parenthesis is the number of subjects for one of the five cancer types.

Cluster ID	Distribution
1	BRCA (57) LUAD (2)
2	LUAD (139)
3	PRAD (136)
4	BRCA (243)
5	KIRC (146)
6	COAD (78)

**Table 4 genes-13-02036-t004:** The distribution of the hierarchical cluster assignments using complete linkage (default settings in R are used) and the gap statistic to select the number of clusters. The gap statistic suggests 7 clusters.

Cluster ID	Distribution
1	BRCA (57) COAD (7) KIRC (42) LUAD (32) PRAD (25)
2	BRCA (52) COAD (8) LUAD (18) PRAD (22)
3	BRCA (84) COAD (17) LUAD (17)
4	COAD (17) LUAD (34) PRAD (40)
5	BRCA (68) KIRC (23) LUAD (18)
6	BRCA (70) COAD (8) KIRC (29) LUAD (22) PRAD (49)
7	KIRC (52) COAD (21)

**Table 5 genes-13-02036-t005:** The distribution of the hierarchical cluster assignments using single linkage (default settings in R are used) and the gap statistic to select the number of clusters. The gap statistic suggests exactly 1 cluster.

Cluster ID	Distribution
1	BRCA (300) COAD (78) KIRC (146) LUAD (141) PRAD (136)

**Table 6 genes-13-02036-t006:** The distribution of the hierarchical cluster assignments using median linkage (default settings in R are used) and the gap statistic to select the number of clusters. The gap statistic suggests exactly 1 cluster.

Cluster ID	Distribution
1	BRCA (300) COAD (78) KIRC (146) LUAD (141) PRAD (136)

**Table 7 genes-13-02036-t007:** The distribution of the hierarchical cluster assignments using k-means (default settings in R are used) and the gap statistic to select the number of clusters. The gap statistic suggests 5 clusters.

Cluster ID	Distribution
1	BRCA (68) COAD (27) KIRC (39) LUAD (42) PRAD (28)
2	BRCA (77) COAD (14) KIRC (31) LUAD (26) PRAD (41)
3	COAD (34) LUAD (1)
4	BRCA (57) COAD (3) KIRC (40) LUAD (46) PRAD (37)
5	BRCA (98) KIRC (36) LUAD (26) PRAD (30)

## Data Availability

The data are publicly available from the UCI data repository mentioned in the manuscript.

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
