# Peer review of "Clustering Gene Expressions Using the Table Invitation Prior"

_genes, 2022, doi:10.3390/genes13112036_

Round 1

Reviewer 1 Report

Harrison, He and Huang developed a framework to cluster Gene Expressions data using a method called Table Invitation Prior (TIP). If I understood well, the main aim of using TIP was to find a method which classify better the overllaped samples to provide a clear interpretation to the users.

They compared TIP with two other methods, EPA and MCLUST. The proposed method was computationally tested using simulations. The results showed that TIP for all kind of data performed better than the others two, in the sense that most samples were classified perfectly in each cluster, with almost none overlapped samples.

Broad comments:

- The sections Materials and Methods (M&M) and Results need to be separated. I did not know when I was reading a M&M or results.

- The M&M need to be more clear with the steps to follow.

- It was not clear for me if the authors also showed the method for real data or just simulated data? In case all the data was simulated, why the authors did not test with real data?

- It was not clear for me in the section of the simulation how was used the univariate multiple change point detection method?

- The numbering lines is not continue in each paragraph.

- I am a little loss with the nomenclature. Sometimes subjects are called by i or sigma_i or x_i or y_i (i.e. lines 44, 48, 53 or equation 20, respectively).

- Authors need to clarify more the differences between methods TIP, EPA, MCLUST.

-Line 88. In this paper vector-variate data are considered and the conjugate Normal-Inverse-Wishart (NIW) prior for the mean and covariance is utilized. Could the authors explain why used this prior distribution for the data?

- Section 3.3. Visualizing the posterior similarity matrix. Need to be more clear. The values of the edges are the posterior probability of what? Need to clarify why small edges are removed. The sentence 'which is the result of removing the maximum number of edges in the graph BÌ„ such that the number of components in the graph is 1' is not clear for me, what they want to say with 'number of components'?

- The Inverse Wishart distribution nomenclature is different in section 4.1. (Simulation Description) and 3.2. (Likelihood Function). It can to create confusion.

- In section 4.1. Simulation Description. Why 5 clusters? Why authors did not do a sensitivity analysis using different cluster sizes?

- Line 110. Subsection 4.2.1. Is this section a scenario? What kind of scenario? I thought that the authors there were done simulations for 5 cluster size. In this scenario I see only 4 and in the section 4.1. (Simulation Description) are described 5.

- Formula 5. What is the meaning of q?

- Line 58. What is t in K(t)?

- Line 58. What is the limit of K? The same number of individuals?

- Line 58. How many subjects there were in each table? Namely, How are distributed the individuals in the restaurant?

- Line 65. Discrete uniform distribution??? Why? Because all individuals have the same probability to be chosen?

Formula 3. Why did the authors choose the median as the measure? In case that it is because of some literature reference, please give it.

Lines 19:21. Bayesian nonparametrics refer to a flexible class of Bayesian priors,and one consequence of such priors in the context of clustering is that an analyst does not need to specify an estimate for the number of clusters. WHY? Furthermore, Bayesian priors what? a probability function?

Line 47. Delta is a “discount” hyperparameter??? Could the authors explain the concept of 'discount'?

Formulas 9 and 11. Is the formula 11 a subset of formula 9?

Formula 13. Is the number of similar subjects to subject r to be always distributed as Poisson in the first change point?

Nomenclature with the posterior probability of the clusters (K_i or K^(i))

After formula 21. ..then the joint posterior for ( μ k , Σ k | y i ) is given by.. Why is this posterior distribution is a "joint" posterior distribution?

Paragraph in line 104. the number of variables d = 3, and the temperature parameter τ = 5??? I understood that d was a distance and tau an hyperparameter.???

Line 120. The 801 samples can be classified into one of 5 classes. What are the classes here?

Author Response

Dear Editor,

We appreciate you and the reviewers for your precious time in reviewing our manuscript and
providing valuable and insightful comments that provide improvements in the revision. The authors have carefully addressed all of them. We hope the revised manuscript will meet your high standards. The authors welcome further constructive
comments if any.
We provide the point-by-point responses below. All modifications in the manuscript have
been highlighted in red.

Sincerely,
Hsin-Hsiung Bill Huang, PhD
[email protected]
Associate Professor,
Department of Statistics and Data Science
University of Central Florida 
\section{Reviewer 1}
\begin{enumerate}
    \item The sections Materials and Methods (M\&M) and Results need to be separated. I did not know when I was reading a M\&M or results. The M\&M need to be more clear with the steps to follow.
    \begin{itemize}
        \item \textbf{Response}:
        The methods and results are in different sections. The proposed method is described in Section 2 and Section 3, the simulation studies are in Section 4 and the real-world gene data application is in Section 5. 
    \end{itemize}
    \item It was not clear for me if the authors also showed the method for real data or just simulated data? In case all the data was simulated, why the authors did not test with real data?
    \begin{itemize}
        \item \textbf{Response}: The proposed method can be applied to both the simulation and real data. Section 4, entitled “4. Simulation Data” refers to applying the method to simulation data whereas the Section 5, entitled “Application: Clustering Gene Expression Data”, refers to a real-world dataset.
    \end{itemize}
    \item It was not clear for me in the section of the simulation how was used the univariate multiple change point detection method?
    \begin{itemize}
        \item \textbf{Response}: The change point detection algorithm is applied to the sorted distances with respect to a subject with index $i$, and the result is utilized in the estimation of $\hat{\tau}_{r,1}$ (see equation 13). The estimate is used to obtain the number subjects that are similar to subject $r$, which follows a Poisson distribution:
\begin{equation*}
\hat{n}_{\tau_{r}} \sim Poi(\hat{\tau}_{r,1}).
\end{equation*}
    \end{itemize}
    \item I am a little loss with the nomenclature. Sometimes subjects are called by $i$ or $sigma_i$ or $x_i$ or $y_i$ (i.e. lines 44, 48, 53 or equation 20, respectively).
    \begin{itemize}
        \item \textbf{Response}:  We agree that our current nomenclature with regard to the term ``subject'' may be confusing. We have added the following to the introduction section. Note that, in practice, an individual subject may correspond to an individual vector, matrix, or higher-order tensor variables. In this work, vectors are considered for the sake of simplicity. In contrast, a ``subject index'' refers to an identifier for a subject. For example, in vector-variate data a subject index $i$ refers to the $i$th row in the provided dataset. The notation $\mathbf{x}_i$ refers to a subject (i.e. vector, matrix, tensor, etc.) whereas the $i$ refers to its index.
    \end{itemize}
    \item Authors need to clarify more the differences between methods TIP, EPA, MCLUST.
    \begin{itemize}
        \item \textbf{Response}: Thank you for pointing this out. We added the following paragraph describing the differences between methods TIP, EPA, MCLUST in Section 6.

        EPA is a prior on partitions and is used for Bayesian clustering. Unlike TIP, the probability of a new cluster in EPA depends on preset hyperparameters (i.e. $\delta$ and $\alpha > -\delta$), which is not data-driven, and it may lead to a bias of number of clusters due to improper hyperparameter values. The main difference between TIP and MCLUST is that TIP is a Bayesian cluster prior which can be incorporated with various types of likelihoods and priors for the parameters in the likelihood. For example, TIP can work with the conjugate prior, the Normal-Inverse-Wishart distribution of for unknown mean and covariance matrix. Nonetheless, MCLUST is based on a mixture model of finite Gaussian likelihoods and using an expectation–maximization (EM) algorithm \cite{dempster1977maximum} for the Gaussian mixture parameter estimation with a preset covariance structure.
    \end{itemize}
    \item In this paper vector-variate data are considered and the conjugate Normal-Inverse-Wishart (NIW) prior for the mean and covariance is utilized. Could the authors explain why used this prior distribution for the data?
    \begin{itemize}
        \item \textbf{Response}: The NIW conjugate prior model is used for a variety of reasons. 1) As a consequence of the Central Limit Theorem, in practice it is not uncommon to find that data are distributed according to a normal distribution \cite{mcelreath2020statistical}. 2) The NIW is a conjugate model which means that the posterior distribution (given a cluster $k$) has a closed-form expression and thus sampling is efficient. This contrasts with other likelihood models that require a Metropolis step where each sample from the posterior distribution may or may not be accepted; consequently the Gibbs sampling algorithm is less efficient and incurs longer runtimes. 
    \end{itemize}
    \item Section 3.3. Visualizing the posterior similarity matrix. Need to be more clear. The values of the edges are the posterior probability of what? 
    \begin{itemize}
        \item \textbf{Response}: The first sentence of Section 3.3 explains this: “The matrix $\bar{\mathbf{B}}$ is an undirected weighted graph where the edge between subject $i$ and subject $j$ is the posterior probability that subject $i$ and subject $j$ are in the same cluster.” We added this sentence in the caption of Figure 4.
    \end{itemize}
    \item Need to clarify why small edges are removed. The sentence 'which is the result of removing the maximum number of edges in the graph $\bar{B}$ such that the number of components in the graph is 1' is not clear for me, what they want to say with 'number of components'?
    \begin{itemize}
        \item \textbf{Response}: Thank you for the comment. The reasons that small edges are removed are described at the beginning of Section 3.3 (see the second, third, and fourth sentences). “A network plot may be used to view  $\bar{\mathbf{B}}$ directly, but the number of edges corresponding to small posterior probabilities may unnecessarily complicate the plot.  Consequently, we show the plot of the graph $\bar{\mathbf{B}}^{(1)}$ which is the result of removing the maximum number of edges in the graph $\bar{\mathbf{B}}$ such that the number of components in the graph is 1. That is, the graph $\bar{\mathbf{B}}^{(1)}$ has minimum entropy among all subgraphs with one component and we call its corresponding network plot the ``one-cluster plot''. The idea is to remove as many connections as possible while still maintaining one component so that the clusters' relationships with each other are revealed. 
    \end{itemize}
    \item The Inverse Wishart distribution nomenclature is different in section 4.1. (Simulation Description) and 3.2. (Likelihood Function). It can to create confusion. 
    \begin{itemize}
        \item \textbf{Response}: The ``inverse Wishart distribution'' terminology used in Section 4.1 refers to the probability distribution. The Normal-Inverse-Wishart distribution is an entirely different probability distribution.
    \end{itemize}
    \item In section 4.1. Simulation Description. Why 5 clusters?
    \begin{itemize}
        \item \textbf{Response}: The purpose of simulation is to examine the performance of the proposed method. Therefore we decide to show the results for 5 clusters as it is naturally more difficult as compared to a smaller number of clusters. 
    \end{itemize}
    \item  Why authors did not do a sensitivity analysis using different cluster sizes?
    \begin{itemize}
        \item \textbf{Response}: Different cluster sizes have been used to evaluate the proposed method in the simulation study. Specifically, the size of each cluster is  20, 25, 30, and 35 vectors, respectively. 
    \end{itemize}
    \item Line 110. Subsection 4.2.1. Is this section a scenario? What kind of scenario? I thought that the authors there were done simulations for 5 cluster size. In this scenario I see only 4 and in the section 4.1. (Simulation Description) are described 5.
    \begin{itemize}
        \item \textbf{Response}: Thank you for pointing this out. We have removed the following:  \textbf{The data consists of 5 clusters whose sizes are approximately $(.30)n$, $(.25)n$, $(.20)n$, $(.15)n$, and $(.10)n$, respectively. In this simulation, $n \in \{100, 1000\}$, the number of variables $d = 3$, and the temperature parameter $\tau = 5$ (see equation 2). }
    \end{itemize}
    \item Formula 5. What is the meaning of q?
    \begin{itemize}
        \item \textbf{Response}: Thank you for pointing this out. We have added the following: $q_{i-1}$ is the number of clusters that are assigned among the first $i-1$ subjects.
    \end{itemize}
    \item Line 58. What is $t$ in K(t)?
    \begin{itemize}
        \item \textbf{Response}: This is answered in line 57: ``the Table Invitation Prior (TIP) is presented in the context of a Gibbs sampler in iteration $t = 1, 2, \ldots, T$.''
    \end{itemize}
    \item Line 58. What is the limit of K? The same number of individuals?
    \begin{itemize}
        \item \textbf{Response}: The lower bound for $K^{(t)}$ is 1 whereas the upper bound is $n$ where $n$ is the number of subjects (i.e. the sample size or the number of gene expressions). 
    \end{itemize}
    \item Line 58. How many subjects there were in each table? Namely, How are distributed the individuals in the restaurant?
    \begin{itemize}
        \item \textbf{Response}: The number of subjects at each table is random and is determined after sampling from the posterior distribution. The distribution of the individuals follows the posterior distribution which is composed of the likelihood model (e.g. the NIW model) and the prior distribution (i.e. the TIP prior).
    \end{itemize}
    \item Line 65. Discrete uniform distribution??? Why? Because all individuals have the same probability to be chosen?
    \begin{itemize}
        \item \textbf{Response}: Yes, all individuals have the same probability of being chosen. In general, there is no reason to prefer sampling a specific subject more than another. However, if, for some reason, there is a reason to prefer sampling specific subjects more than others then a different discrete distribution may be used by the analyst.
    \end{itemize}
    \item Formula 3. Why did the authors choose the median as the measure? In case that it is because of some literature reference, please give it.
    \begin{itemize}
        \item \textbf{Response}: Thank you for pointing this out. We added the following to the paper: ``Following the approach taken in \cite{gretton2012kernel}, the hyperparameter $\tau$ is set ...."

        Also, we added the following to the paper to provide a more detailed explanation: 

        The choice of the median is heuristic; however there is a justification. Equation (3) implies that that 
\begin{equation*}
\lim\limits_{d_{ij} \xrightarrow{} \infty} \exp(\frac{-d_{ij}}{\Tilde{d}}) = 0,
\end{equation*}
\begin{equation*}
\lim\limits_{d_{ij} \xrightarrow{} 0} \exp(\frac{-d_{ij}}{\Tilde{d}}) = 1,
\end{equation*}
and 
\begin{equation*}
\lim\limits_{d_{ij} \xrightarrow{} \Tilde{d}} \exp(\frac{-d_{ij}}{\Tilde{d}}) = \exp(-1).
\end{equation*}
Consequently, similarity values corresponding to subject pairs whose distances that are significantly larger than the median go to zero whereas subject pairs that are very close to each other are closer to 1. Subject pairs whose distance from each other are relatively commonplace have smaller values relative to subject pairs that are close to each other. 

    \end{itemize}
    \item Lines 19:21. Bayesian nonparametrics refer to a flexible class of Bayesian priors, and one consequence of such priors in the context of clustering is that an analyst does not need to specify an estimate for the number of clusters. WHY? Furthermore, Bayesian priors what? a probability function?
    \begin{itemize}
        \item \textbf{Response}: Thank you for pointing this out. We added the following to the paper: Bayesian nonparametrics refer to a flexible class of prior distributions that may be used in a variety of settings including clustering. In the context of clustering, the use of such priors means that an analyst does not need to specify an estimate for the number of clusters since the number of clusters is modeled as a random variable.
    \end{itemize}
    \item Line 47. Delta is a “discount” hyperparameter??? Could the authors explain the concept of 'discount'?
    \begin{itemize}
        \item \textbf{Response}: Thank you for pointing this out. We have added the following to the paper to give more information about the discount parameter. The discount parameter is specific to EPA and its purpose is to incorporate information about the number of clusters in a previous iteration when computing the probability of a new cluster in each iteration.
    \end{itemize}
    \item Formulas 9 and 11. Is the formula 11 a subset of formula 9?
    \begin{itemize}
        \item \textbf{Response}: No, formula (11) is not a subset of formula (9). Formula (9) refers to a set of similarity values whereas Formula (11) refers to a set of subject indices.  
    \end{itemize}
    \item Formula 13. Is the number of similar subjects to subject $r$ to be always distributed as Poisson in the first change point?
    \begin{itemize}
        \item \textbf{Response}: Yes, the number of similar subjects to subject $r$ is always distributed as Poisson with the rate parameter equal to the first change point value. 
    \end{itemize}
    \item Nomenclature with the posterior probability of the clusters ($K_i$ or $K^{(i)}$)
    \begin{itemize}
        \item \textbf{Response}: Thank you for pointing this out. We have changed the notation to be $K^{(t)}$. 
    \end{itemize}
    \item After formula 21, then the joint posterior for $\mu_k, \boldsymbol{\Sigma}_k \mid y_i$ is given by ... Why is this posterior distribution is a "joint" posterior distribution?
    \begin{itemize}
        \item \textbf{Response}: This is the posterior distribution of both $\mu_k$ and $\Sigma_k$, hence the term ``joint posterior distribution''.  
    \end{itemize}
    \item Paragraph in line 104. the number of variables d = 3, and the temperature parameter $\tau$ = 5??? I understood that d was a distance and tau an hyperparameter.
    \begin{itemize}
        \item \textbf{Response}: Thank you for pointing this out. The $\tau = 5$ was left in by mistake from a previous draft, and the value for $\tau$ is given by equation (3). Also, we have changed $d$ to be p in order to avoid any confusion. 
    \end{itemize}
    \item Line 120. The 801 samples can be classified into one of 5 classes. What are the classes here?
    \begin{itemize}
        \item \textbf{Response}: Thank you for pointing this out. We have added the following to the paper.  The 5 classes correspond to different types of cancer: BRCA, COAD, KIRC, LUAD, and PRAD.
    \end{itemize}
\end{enumerate}

Reviewer 2 Report

The authors presented an interesting clustering method named Table Invitation Prior (TIP). As a clustering method, the method is computationally feasible, has a nice interpretation within the Bayesian framework, and does not need users to specify the number of clusters. 

I feel that the method is a novel contribution, yet, the authors fail to sufficiently demonstrate its improvement over existing methods. I have a few comments on the evaluation: 

1) The numerical example presented with four clusters that are well separated. Essentially no algorithm is needed and eye balling is sufficient. The authors should consider scenarios where the cluster may be partially overlapping and evaluate if different methods can separately them apart. 

2) the selection of the number clusters needs to be evaluated as well in more realistic scenarios. Classical methods such as hierarchical cluster do not automatically select cluster themselves, but there are many methods that aid the selection of such clusters. Examples include the Gap statistic by Tishirani 2002. Can the authors compare their method with existing approaches? 

Author Response

Dear Editor,

We appreciate you and the reviewers for your precious time in reviewing our manuscript and
providing valuable and insightful comments that provide improvements in the revision. The authors have carefully addressed all of them. We hope the revised manuscript will meet your high standards. The authors welcome further constructive
comments if any.
We provide the point-by-point responses below. All modifications in the manuscript have
been highlighted in red.

Sincerely,
Hsin-Hsiung Bill Huang, PhD
[email protected]
Associate Professor,
Department of Statistics and Data Science
University of Central Florida 

\section{Reviewer 2}
\begin{enumerate}
    \item The authors presented an interesting clustering method named Table Invitation Prior (TIP). As a clustering method, the method is computationally feasible, has a nice interpretation within the Bayesian framework, and does not need users to specify the number of clusters.
    \item I feel that the method is a novel contribution, yet, the authors fail to sufficiently demonstrate its improvement over existing methods. I have a few comments on the evaluation: 1)    The numerical example presented with four clusters that are well separated. Essentially no algorithm is needed and eye balling is sufficient. The authors should consider scenarios where the cluster may be partially overlapping and evaluate if different methods can separately them apart. 
    \begin{itemize}
        \item \textbf{Response}: Thank you for pointing this out. We have entitled the first simulation as a ``Well Separated Clusters'', and we have added a second simulation (please see Section 4.2.2) called ``Overlapped Clusters''.
    \end{itemize}
    \item The selection of the number clusters needs to be evaluated as well in more realistic scenarios. Classical methods such as hierarchical cluster do not automatically select cluster themselves, but there are many methods that aid the selection of such clusters. Examples include the Gap statistic by Tishirani 2002. Can the authors compare their method with existing approaches? 
    \begin{itemize}
        \item \textbf{Response}: Thank you for pointing this out. We have compared TIP, EPA, and MCLUST with hierarchical clustering (various linkages) where the number of clusters are chosen via the gap statistic. Please see Section 4.2.1, Section 4.2.2, and Section 5.  
    \end{itemize}
\end{enumerate}

Round 2

Reviewer 2 Report

The authors adequately addressed my comments. I am supportive of publication.